# Transiently antigen-primed B cells return to naive-like state in absence of T-cell help

Jackson S. Turner[1], Matangi Marthi[1], Zachary L. Benet[1] & Irina Grigorova[1]

The perspective that naive B-cell recognition of antigen in the absence of T-cell help causes cell death or anergy is supported by *in vivo* studies of B cells that are continuously exposed to self-antigens. However, intravital imaging suggests that early B-cell recognition of large foreign antigens may be transient. Whether B cells are tolerized or can be recruited into humoural immune responses following such encounters is not clear. Here we show that in the presence of T-cell help, single transient antigen acquisition is sufficient to recruit B cells into the germinal centre and induce memory and plasma cell responses. In the absence of T-cell help, transiently antigen-primed B cells do not undergo apoptosis *in vivo*; they return to quiescence and are recruited efficiently into humoural responses upon reacquisition of antigen and T-cell help.

[1] Department of Microbiology and Immunology, University of Michigan Medical School, 1150W. Medical Center Drive, Ann Arbor, Michigan 48109, USA. Correspondence and requests for materials should be addressed to I.G. (email: igrigor@umich.edu).

In accord with the two-signal model of lymphocyte activation proposed by Bretscher and Cohn, B cells must receive a primary activating signal through antigen-dependent crosslinking of B-cell receptors (BCRs) and a second signal from activated cognate helper T (Th) cells to get recruited into long-term, high-affinity humoral immune responses[1–4]. In secondary lymphoid organs, after antigen binding triggers BCR signalling, activated B cells internalize the antigen by receptor-mediated endocytosis and then undergo a series of phenotypic and migrational changes[5]. Within 6 h, activated B cells upregulate expression of the costimulatory molecule CD86 (ref. 6) and present proteolysed antigenic peptides in complex with MHCII for recognition by cognate Th cells[7]. They also upregulate CCR7, a G-protein coupled receptor specific for the T zone chemokines CCL19 and CCL21 and migrate from B-cell follicles to the B-T zone border and interfollicular area where they can interact with cognate Th cells[8–10]. After acquisition of T-cell help, B cells undergo proliferation and differentiate into short-lived plasma cells (PCs) and germinal centre (GC) B cells. Although B cells with a wide range of BCR affinities to antigen are recruited into the T-dependent B-cell response in vivo[11], the clones with lower antigen binding strengths generate reduced short-lived PC responses, due to limited expansion of plasmablasts[12,13], and populate GCs less when faced with interclonal competition for T-cell help[14]. Within GCs, B cells then undergo somatic hypermutation of the BCR and compete for antigen deposited on follicular dendritic cells (FDC) and help from follicular Th cells in a process that drives GC B cells' affinity maturation and formation of high-affinity class-switched memory cells and long-lived PCs[15]. Although many studies have focused on the cellular, molecular and chemotactic mechanisms regulating B-cell response initiation and GCs, how temporal dynamics of antigen and T-cell help acquisition by B cells affect engagement with T-cell-dependent humoral responses in vivo is unclear.

The dynamics of B cells' exposure to foreign antigen in vivo depend on multiple factors, including the antigen's physical properties, route of entry and formation of immune complexes. Small antigens (for example, toxins, proteolysed pathogen fragments) quickly permeate B-cell follicles, whereas initially, large antigens are often restricted to the subcapsular and medullary sinuses and interfollicular areas of the lymph nodes[16]. By 2-photon imaging it has been shown that, during initiation of the B-cell response, naive antigen-specific B cells can transiently approach these regions (for a few minutes to a few tens of minutes), acquire the large antigens and then return to B-cell follicles[17–19]. However, due to technical limitations, the precise history of antigen acquisition by these cells and their fate has not been possible to study.

A previous ex vivo study of B-cell signalling and transcriptional regulation suggests that a single round of BCR signalling may be sufficient to prime B cells for acquisition of T-cell help. However, it also suggests that survival of transiently antigen-primed B cells in the absence of T-cell help is compromised[20]. This observation is consistent with Polly Matzinger's hypothesis that to maintain tolerance, B cells that acquire antigen but not T-cell help must die[21]. Supporting this proposal, multiple in vivo studies demonstrated that B cells that continuously acquire self-antigen undergo apoptosis or anergy[22,23]. However, the fate in vivo of B cells transiently exposed to antigen is unclear, both with respect to induction of tolerance and recruitment into T-cell-dependent humoral immune responses. Here we show that transient antigen acquisition enables B-cell participation in GC, memory B cell and PC responses when T-cell help is available and allows B cells to return to a naive-like state when it is not, rather than undergo anergy or apoptosis.

## Results

**Antigen-primed B cells are recruited into humoural responses.** To determine the fate of B cells after a single transient acquisition of antigen in vivo we utilized the following approach. BCR transgenic (Ig-Tg) HyHEL10 B cells specific for hen egg lysozyme (HEL)[24] were pulsed ex vivo for 5 min with HEL fused to ovalbumin (HEL-OVA), unbound antigen was washed off, and the cells transferred into recipient mice, which had been pre-injected with transgenic OTII Th cells specific to peptide ova$_{323-339}$ in I-A$^b$ (ref. 25) and pre-immunized with ovalbumin (OVA) in complete Freund's adjuvant (CFA) (Fig. 1a). While HEL-OVA-primed B cells could not reacquire cognate HEL antigen in vivo, they could digest pre-acquired OVA, present OVA-derived peptides and make cognate interactions with activated OVA-specific Th cells. In inguinal lymph nodes (ILNs) of OVA-immunized recipient mice, antigen-primed B cells underwent proliferation and were transiently recruited into GCs (Fig. 1b–e and Supplementary Fig. 1). They also differentiated into memory B cells (CFSE$^{low}$ GL7$^{low}$ CD38$^{high}$, both IgD$^{pos}$ and class-switched) and PCs, and generated a modest class-switched antibody (Ab) response, predominantly of the IgG$_1$ isotype (Fig. 1f–i and Supplementary Fig. 1). Recruitment of Ig-Tg cells into the B-cell response was dependent on cognate interactions with activated Th cells, as Ig-Tg B cells pulsed with HEL did not form GCs or PCs in OVA immunized recipient mice (Fig. 1e,g, day 6, hashed bars), similarly to HEL-OVA pulsed Ig-Tg cells in unimmunized control mice (Fig. 1a,e,g). To verify these results were not an artifact of high-affinity antigen, Ig-Tg B cells were primed with duck egg lysozyme (DEL) antigen conjugated to OVA, which has a more physiologic, 10$^3$-fold lower affinity to Ig-Tg BCRs compared to HEL[26]. Comparable participation of Ig-Tg cells in the B-cell response was observed following priming with HEL-OVA and DEL-OVA (Fig. 1e–g, i–l). Together, these data suggest that a single transient antigen acquisition may be sufficient to enable B cells' participation in the T-dependent B-cell response in vivo.

Interestingly, initial recruitment of antigen-pulsed Ig-Tg cells into the GC and memory responses in HEL-OVA and DEL-OVA immunized mice, in which Ig-Tg cells could reacquire cognate antigen in vivo, was not significantly different from OVA immunized mice (Fig. 1e,f,j,k). However, we observed a trend for an increased Ig-Tg PC response in HEL-OVA and DEL-OVA immunized recipient mice (Fig. 1g,l, day 8), which was reflected in elevated IgG$_1^a$ antibody titers (Fig. 1i).

Ig-Tg B cells were recruited into the T-dependent B-cell response after pulsing with DEL-OVA concentrations ranging from 0.005 to 50 μg ml$^{-1}$ (Fig. 2a–h). Of interest, we found that Ig-Tg B cells formed GC, memory and PCs even at antigen doses suboptimal for B-cell activation (based on their failure to induce CD69 and CD86 upregulation or surface IgM downregulation, Fig. 2a–d). However, significantly decreased accumulation of PCs was observed at DEL-OVA concentrations below 0.05 μg ml$^{-1}$, and GC and memory cells at 0.005 μg ml$^{-1}$, consistent with previous studies suggesting greater dependence of PC output on responding B cells' affinity for antigen (Fig. 2e–h)[12–14]. Moreover, B cells were recruited into the T-cell-dependent response when the estimated number of antigen-pulsed Ig-Tg B cells that entered draining LNs was within the range of reported frequencies (10$^{-5}$–10$^{-4}$) of endogenous B cells specific to a particular antigen[27,28] and could be supported by the endogenous OVA-specific Th repertoire (Fig. 2i–l). Based on these data, we conclude that when T-cell help is not limiting, single transient acquisition of antigen over a wide range of doses may be sufficient for the initial recruitment of rare antigen-specific B cells into the T-dependent response.

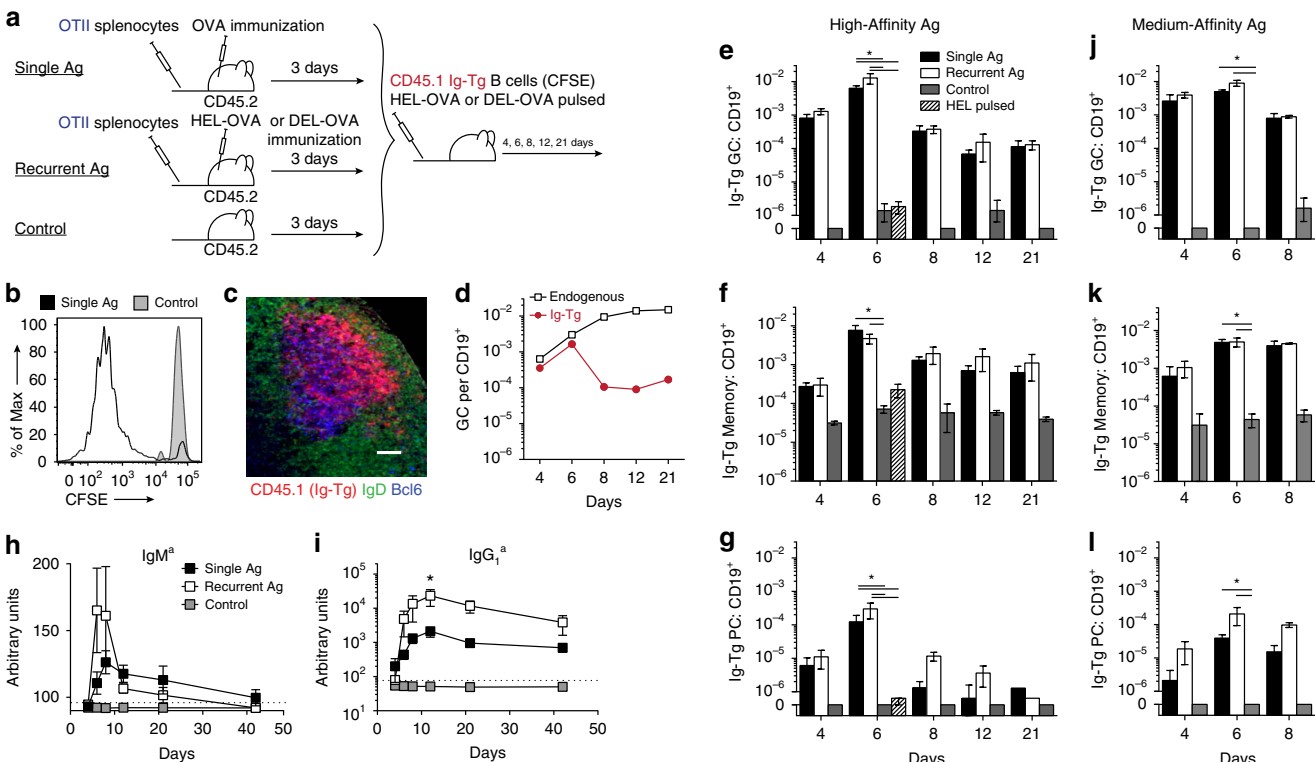

**Figure 1 | A single exposure to antigen enables B cell participation in the T-dependent humoural immune response *in vivo*. (a)** Experimental outline. CFSE-labelled HyHEL10 Ig-Tg B cells were pulsed *ex vivo* for 5 min with 50 μg ml$^{-1}$ HEL-OVA or DEL-OVA, washed and transferred into recipient mice pre-injected with OTII Th cells and s.c. preimmunized with OVA, HEL-OVA or DEL-OVA in CFA or into unimmunized control mice. **(b–d)** Recruitment of HEL-OVA pulsed Ig-Tg B cells into the B-cell response in draining ILNs of mice immunized with OVA. **(b)** Proliferation of antigen-pulsed Ig-Tg cells 4 days post transfer (d.p.t.) in OVA-immunized (Single Ag) or unimmunized (Control) recipient mice. **(c)** Confocal micrograph of IgD$^{low}$Bcl6$^+$ GC at 6 d.p.t. Scale bar = 70 μm. **(d)** Kinetics of endogenous (white boxes) and antigen-pulsed Ig-Tg (red circles) B cells' participation in GC response. Representative of $n = 3$ **(b,d)** or $n = 2$ **(c)** independent experiments. **(e–l)** Kinetics of HEL-OVA (high-affinity antigen, **e–i**), DEL-OVA (medium-affinity antigen, **j–l**) or HEL (1 μg ml$^{-1}$, hashed bar, **e–g**, day 6 only) pulsed Ig-Tg B cells' participation in the GC, memory, PC and antibody response in mice immunized with OVA (black and hashed symbols), HEL-OVA (**e–i**, white symbols), DEL-OVA (**j–l**, white symbols) in CFA or in unimmunized mice (grey symbols). See also Supplementary Fig. 1. Data are from ILNs, $n = 3$ independent experiments, three mice per condition, except for day 6, which is from $n = 7$ independent experiments, nine mice per condition (**e–g**, **j–l**, HEL-OVA and DEL-OVA pulsed) or two independent experiments; six mice per condition (**e–g**, HEL pulsed). Data shown as mean ± s.e.m. *$P < 0.05$ (Kruskal–Wallis test with Dunn's post-test). **(h,i)** Serum αHEL IgM$^a$ (**h**) and the predominant class-switched IgG$_1^a$ (**i**) antibody titers. Dotted line represents ELISA limit of detection. Data are from $n = 4$ independent experiments with 4–5 mice per timepoint. Data shown as mean ± s.e.m. *$P < 0.05$ (unpaired two-tailed Student's *t*-test between single and recurrent Ag).

**Antigen-primed B cells survive in the absence of T-cell help**. To study the fate of antigen-primed B cells in the absence of T-cell help, we utilized Ig-Tg MD4 B cells, which constitute about 95% of total B cells in MD4 mice, allowing for more straightforward enumeration following adoptive transfer. MD4 B cells express a Tg BCR similar to that of HyHEL10 B cells and similarly bind DEL-OVA (Fig. 2b and Supplementary Fig. 2c), but cannot undergo class-switching[29]. To address the fate of antigen-primed B cells in the absence of immediate T-cell help *ex vivo* and *in vivo*, antigen-pulsed Ig-Tg B cells were either co-cultured with control naive B cells or co-transferred into unimmunized recipient mice (Fig. 3a). Ig-Tg MD4 B cells were pulsed with a high dose of DEL-OVA (50 μg ml$^{-1}$, Fig. 2a–d) or DEL-OVA conjugated to E-alpha peptide (Eα) which is recognized in complex with MHCII by the Y-ae antibody and can be used to monitor antigen-derived peptide presentation[30]. While previous work and our studies indicate that antigen-primed B cells undergo rapid apoptosis when cultured *ex vivo* (Fig. 3b)[20], we observed no substantial decrease in the numbers of antigen-primed B cells within 3 days of their transfer into unimmunized recipient mice (Fig. 3c). A minor population (<7%) of antigen-primed Ig-Tg B cells proliferated in recipient

mice (Supplementary Fig. 2a). To avoid the confounding effect of proliferation, quantitative analysis of B-cell numbers was performed on the unproliferated fraction of Ig-Tg cells normalized to cotransferred naive control cells, which did not proliferate. This analysis indicated no progressive apoptosis of the antigen-pulsed B cells *in vivo* (Supplementary Fig. 2a,b). The survival of antigen-primed B cells *in vivo* was independent of cognate or noncognate interactions with Th cells as indicated by similar persistence of antigen-primed B cells in αβ T-cell-deficient TCRα$^{-/-}$ recipient mice (Fig. 3d). Of note, similar proliferation of a small fraction of DEL-OVA-primed Ig-Tg B cells was observed in TCRα$^{-/-}$ recipient mice, indicating that proliferation was T-independent (Supplementary Fig. 2a). In contrast to B cells primed with DEL-OVA only once, continuous re-exposure of B cells to DEL-OVA in the absence of T-cell help led to their progressive loss *in vivo* (Fig. 3e), consistent with previous reports of MD4 B cells transferred into HEL-expressing recipient mice[31,32].

To test how antigen valency affects B-cell survival *in vivo*, we first characterized the ability of various forms of Ig-Tg B-cell antigens to downregulate BCRs *ex vivo*. DEL-OVA induced a similar amount of BCR crosslinking and receptor occupancy in

Ig-Tg B cells as moderately multivalent DEL (mDEL) generated by oligomerization of biotinylated DEL with streptavidin (Supplementary Fig. 2c,d). Both of these antigens induced Ig-Tg BCR crosslinking and internalization substantially more efficiently than monovalent forms of either HEL or DEL, but less efficiently than very highly multivalent polystyrene beads coated

with DEL (sphDEL) (ref. 33; Supplementary Fig. 2c,d). Then, to determine whether antigen valency alters B-cell survival following transient antigen acquisition, Ig-Tg B cells were pulsed with soluble antigen (sHEL), moderately multivalent antigen (DEL-OVA or mDEL), or highly multivalent antigen (sphDEL) and their persistence in vivo was monitored for 7 days (Fig. 3f). Axial, brachial, cervical, inguinal and mesenteric LNs were collected in addition to spleens to account for recirculation of transferred cells. Although transient exposure to soluble or moderately multivalent antigens did not result in B-cell disappearance from the secondary lymphoid organs within this time frame, transient exposure to highly multivalent antigen had a negative effect on B-cell survival, a trend observed as early as 12 h post transfer, with a significant decline in B-cell number by 7 days (Fig. 3f and Supplementary Fig. 2e). The observed decline in unproliferated sphDEL-pulsed B-cell numbers was not due to their increased proliferation in vivo (Fig. 3f and Supplementary Fig. 2f).

**Antigen-primed B cells return to a naive-like state.** As we detected no significant apoptosis of B cells transiently primed with DEL-OVA in the absence of T-cell help in vivo, we asked whether these B cells remained activated and capable of receiving T-cell help. At 12 h after their transfer into unimmunized recipient mice, antigen-primed B cells in the spleen were located predominantly at the borders between B-cell follicles and T-cell zones and had downregulated their surface IgM$^a$ BCRs (Fig. 4a,c,f,g and Supplementary Fig. 3a). By 24 h, the cells re-upregulated their BCRs, downregulated surface expression of CCR7 receptors, and started to relocalize back into B-cell follicles (Fig. 4b–g and Supplementary Fig. 3a). Consistent with B-cell inactivation, progressive downregulation of surface CD86 molecules and MHCII/antigenic Eα complex presentation by antigen-primed B cells were observed between 24 and 72 h (Fig. 4h–k and Supplementary Fig. 3b). CD86 and CCR7 downregulation at 3 and 7 days post transfer was also observed following transient Ig-Tg B-cell priming with sHEL, mDEL and sphDEL. To determine whether the kinetics of B-cell inactivation may depend on the amount of antigen initially acquired, Ig-Tg B cells were pulsed ex vivo with various doses of DEL-OVA-Eα, and then characterized in vivo 6 and 24 h later. Pulsing Ig-Tg B cells with concentrations of DEL-OVA-Eα between 0.025 and 2.5 μg ml$^{-1}$ resulted in a linear increase in surface presentation of antigenic peptide (Fig. 4l,m). Interestingly, as Ig-Tg B cells primed with all concentrations of DEL-OVA-Eα had decreased presentation of I-A$^b$/ Eα by 24 h (Fig. 4n), those primed with lower amounts of antigen downregulated CD86 more quickly, suggesting that duration of B-cell activation may depend on the dose of initially acquired antigen (Fig. 4o).

**Primed B cells are receptive to T-cell help for 24–48 h.** To address the ability of antigen-primed B cells to receive delayed

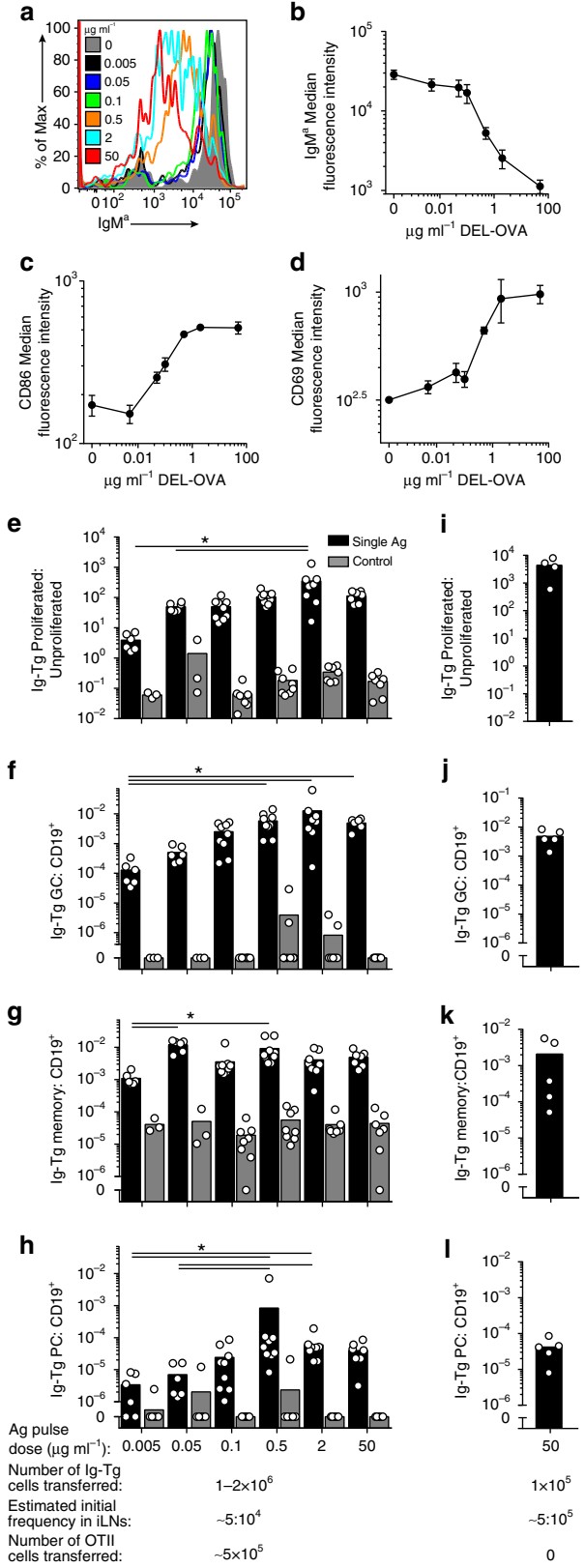

**Figure 2 | A broad range of antigen doses enables recruitment of antigen-primed B cells into T-dependent response in vivo.** (a–d) Ig-Tg B cells' surface IgM$^a$ (BCR) (a,b), CD86 (c) and CD69 (d) expression 6 h following ex vivo pulsing with the indicated doses of DEL-OVA and transfer into naive recipient mice. Spleens analysed. Data are representative of (a) or from three independent experiments, shown as mean ± s.e.m. (b–d). (e–l) Proliferation and formation of GC, memory and PCs by the indicated number of Ig-Tg B cells pulsed ex vivo with the indicated doses of DEL-OVA and transferred into OVA-immunized mice (black bars) or into control unimmunized mice (grey bars) with (e–h) or without (i–l) OTII Th cells. ILNs analysed at 6 d.p.t. Each dot represents one mouse and bars correspond to mean values. n = 2–5 independent experiments. *P < 0.05 (Kruskal–Wallis test with Dunn's post-test).

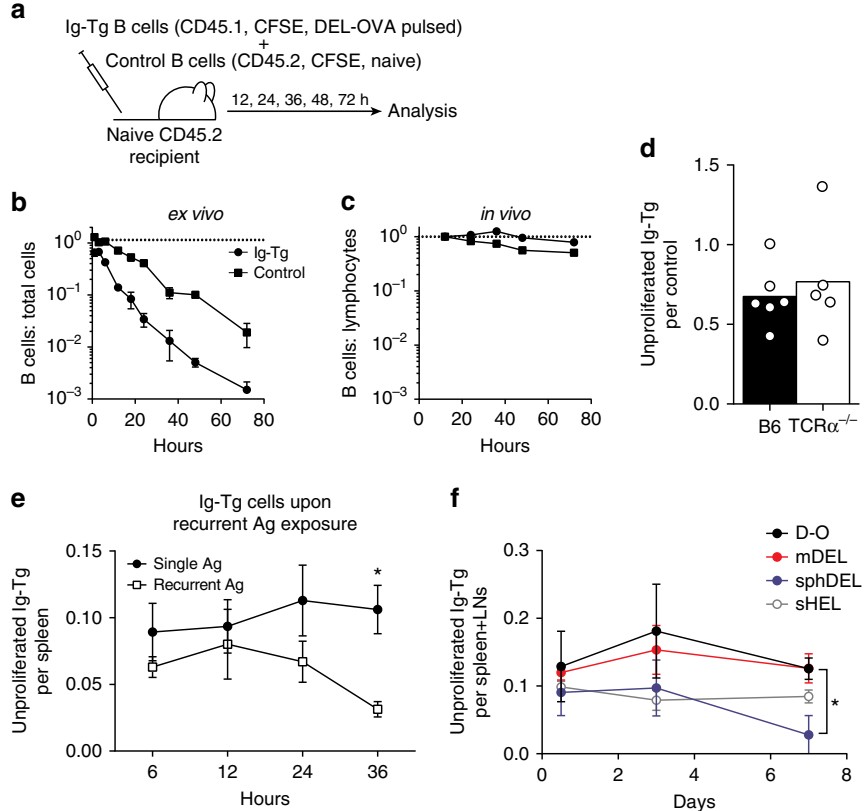

**Figure 3 | Antigen-primed B cells do not undergo apoptosis in the absence of T-cell help in vivo.** (**a**) Experimental outline for **c** and **d**. CFSE-labelled MD4 Ig-Tg B cells were pulsed ex vivo with $50 \mu g \, ml^{-1}$ DEL-OVA or DEL-OVA-Eα and co-transferred with CFSE-labelled naive control B cells into unimmunized mice, then analysed after various periods of time. (**b,c**) Survival of antigen-pulsed Ig-Tg and control B cells in the absence of T-cell help when cultured ex vivo (**b**, normalized to 0 h, $n = 3$ independent experiments) or in vivo in spleens of recipient mice (**c**, normalized to 12 h, $n = 5$ independent experiments). Data shown as mean ± s.e.m. (**d**) Ratios of unproliferated Ig-Tg B cells to control B cells in the spleens of recipient wild type (B6) and TCRα$^{-/-}$ mice at 72 h.p.t. Data are normalized to the injected ratios of Ig-Tg to control B cells. Each symbol represents one mouse, bars at mean. $n = 2$ independent experiments. (**e**) Time-course analysis of DEL-OVA-pulsed ($50 \mu g \, ml^{-1}$) unproliferated Ig-Tg B-cell numbers per spleen in unimmunized recipient mice injected (white dots) or not (black dots) with $50 \mu g$ DEL-OVA i.v. at 0, 12 and 24 h, normalized to the number of injected Ig-Tg cells. Data shown as mean ± s.e.m. $n = 2$ independent experiments with six mice. *$P < 0.05$ (two-tailed Mann–Whitney test). (**f**) Time-course analysis of DEL-OVA ($50 \mu g \, ml^{-1}$), mDEL ($50 \mu g \, ml^{-1}$), sphDEL ($4 \times 10^9$ spheres $ml^{-1}$, $0.17 \mu g \, ml^{-1}$ DEL) and sHEL ($1 \mu g \, ml^{-1}$) pulsed unproliferated Ig-Tg B-cell numbers per combined spleen and LNs in unimmunized recipient mice, normalized to the number of injected Ig-Tg cells. Data shown as mean ± s.e.m. $n = 2$ independent experiments with four mice. *$P < 0.05$ between DEL-OVA pulsed and sphDEL pulsed Ig-Tg B cells (Kruskal–Wallis test with Dunn's post-test).

T-cell help, Ig-Tg B-cells pulsed with a high dose of antigen were transferred from primary recipients into OVA-immunized secondary recipient mice at various times, and their proliferation was assessed (Fig. 5a). As expected based on the kinetics of B-cell inactivation, the majority of Ig-Tg B cells underwent proliferation when T-cell help was available 12 h following antigen priming. By 24 h, only half of the cells could engage into the cell cycle, and by 48 h most of the cells were unresponsive to T-cell help (Fig. 5b–e). These data suggest that after transient acquisition of antigen, B cells have a 1–2-day window (or possibly less for lower doses of acquired antigen) for acquisition of T-cell help before they return to a quiescent state.

**Primed and inactivated B cells can re-enter humoral responses.** To determine whether antigen-primed B cells that underwent inactivation were anergic or could reacquire antigen and T-cell help and mount a productive humoral response in vivo, secondary recipient mice with inactivated antigen-primed Ig-Tg B cells were reimmunized with DEL-OVA (Fig. 6a). Three days after DEL-OVA immunization the majority of previously inactivated Ig-Tg B cells underwent proliferation (Fig. 6b, Supplementary Fig. 4a) and by 6–7 days post immunization

generated strong GC and PC responses, comparable to those of naive Ig-Tg B cells, which had not previously encountered antigen (Fig. 6c,d). To eliminate the possibility that the GC and PC responses were predominantly mounted by a small fraction of antigen-exposed Ig-Tg cells that underwent T-independent proliferation, $3–7 \times 10^3$ unproliferated, antigen-primed and then inactivated or naive Ig-Tg B cells were FACS-sorted from primary recipient mice 5 days after transfer and injected into secondary recipient mice, wherein their recruitment into the B cell response upon DEL-OVA administration was assessed. Unproliferated antigen-primed and then inactivated and naive Ig-Tg B cells generated comparable GC and PC responses (Fig. 6e,g). Of note, under these experimental conditions, the magnitudes of the induced GC and PC responses were sensitive to the number of antigen-responsive B cells, suggesting that antigen-primed then inactivated and naive B cells generated GC and PC responses with similar efficiency (Fig. 6e–h). While the possibility cannot be ruled out that some Ig-Tg B cells remained residually activated and contributed to the observed GC and PC responses, analysis of the sorted cells at the time of transfer suggested these cells would be very rare, as no more than 7% of the antigen-experienced population of Ig-Tg cells expressed

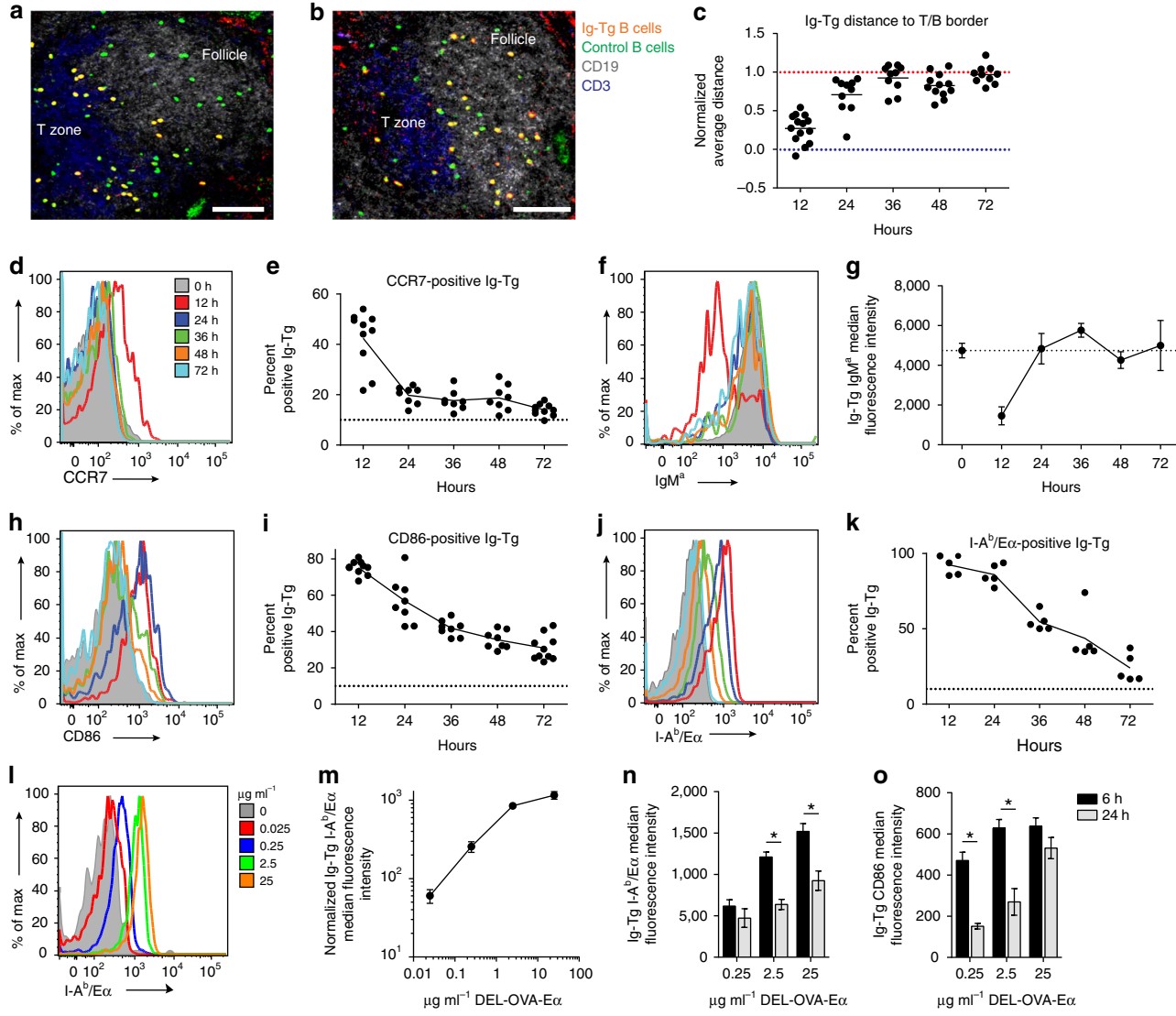

**Figure 4 | In the absence of T-cell help antigen-primed B cells return to a quiescent state *in vivo*.** (**a–k**) Time-course analysis of DEL-OVA or DEL-OVA-Eα-pulsed (50 μg ml$^{-1}$) MD4 Ig-Tg and control B cells' localization (**a–c**) and surface marker expression (**d–k**) in the spleens of unimmunized recipient mice. For experimental set-up see Figure 3a. (**a,b**) Representative confocal micrographs of spleen sections at 12 (**a**) and 24 (**b**) h.p.t of antigen-pulsed Ig-Tg cells. T/B border is at interface between T zone (blue) and follicle (white). Scale bars = 70 μm. (**c**) Average distance of Ig-Tg cells to T/B border, normalized to average distance to the border of randomly distributed points in the follicle. Each dot represents one follicle and adjacent T/B border, line at mean. Blue dotted line corresponds to cell distribution along T/B border, red dotted line to random distribution in the follicles. See Supplementary Fig. 3a. n = 3 independent experiments. (**d,f,h,j**) Representative histograms of Ig-Tg B cells' surface expression of CCR7 (**d**), IgM$^a$ BCR (**f**), CD86, (**h**) and I-A$^b$/Eα (**j**) at the indicated times after transfer into recipient mice. (**e,i,k**) Percent of Ig-Tg cells staining positive for CCR7 (**e**), CD86 (**i**) and I-A$^b$/Eα (**k**). Positive gates defined at fluorescence brighter than 90% of control cells (dotted line at 10%), see Supplementary Fig. 3b. (**g**) IgM$^a$ Median fluorescence intensity. n = 5 (**d,e,h,i**), n = 3 (**j,k**) and n = 2 (**f,g**) independent experiments. Each point corresponds to one mouse (**e,i,k**) or is shown as mean ± s.e.m. for six mice (**g**). (**l–o**) MD4 Ig-Tg B cells' antigenic peptide presentation (**l–n**) and CD86 surface expression (**o**) after pulsing with the indicated concentrations of DEL-OVA-Eα and transfer into naive recipient mice. Surface I-A$^b$/Eα representative histogram (**l**) and median fluorescence intensity normalized to mock-pulsed cells (**m**) 6 h after transfer. Median fluorescence intensity of surface I-A$^b$/Eα (**n**) and CD86 (**o**) at 6 and 24 h after transfer. n = 3 independent experiments, five mice per timepoint, shown as mean ± s.e.m. (**m–o**). *$P < 0.05$ (unpaired two-tailed Student's *t*-test).

higher amounts of CD86 than naive control cells (Supplementary Fig. 4b).

## Discussion

In summary, the findings described above suggest that transient foreign antigen acquisition by B cells can be sufficient for their recruitment into the T-dependent humoral response and generation of GCs, memory B cells and PCs *in vivo*. Previous

*ex vivo* analysis of intracellular signalling and transcriptional regulation in B cells suggested that transient exposure to antigen may be sufficient to prime B cells to receive T-cell help[20]. Consistent with that, we find that *in vivo* in the presence of T-cell help, B cells transiently primed with moderately multivalent antigen undergo proliferation and are recruited into the GC and class-switched memory response. They also generate a modest PC and class-switched antibody response. We observe similar results regardless of whether the numbers of antigen-pulsed

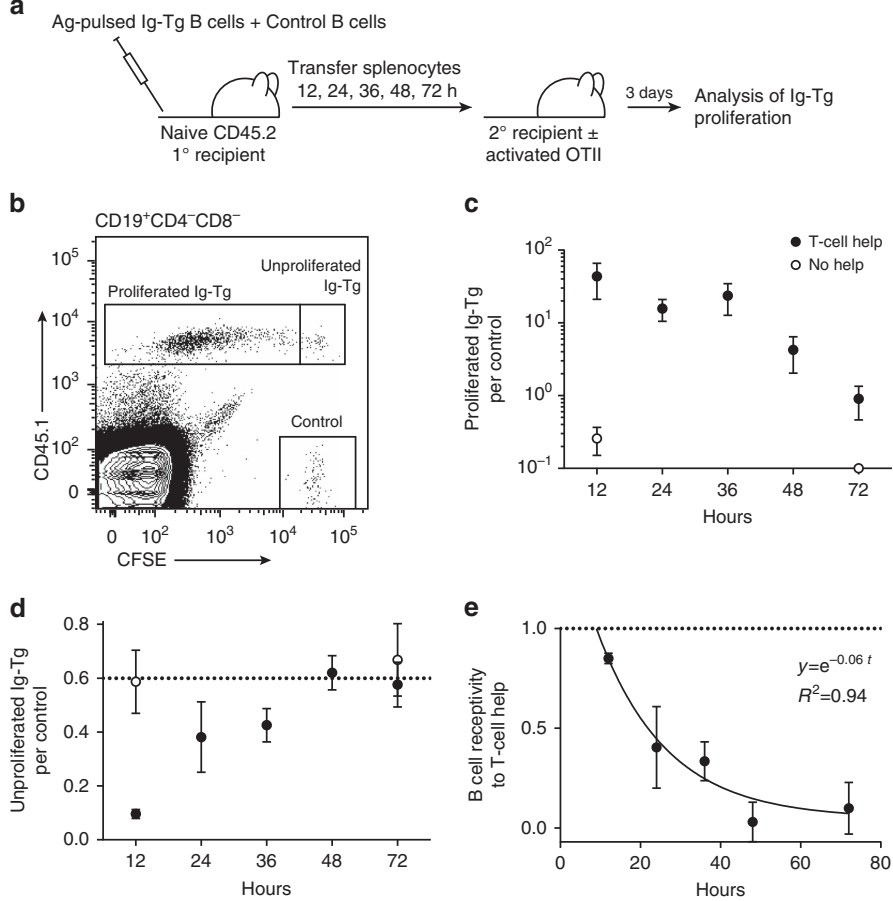

**Figure 5 | B cells lose receptivity to T-cell help in parallel with phenotypic inactivation.** Time-course analysis of antigen-pulsed Ig-Tg B cells' ability to receive T-cell help and undergo proliferation *in vivo*. (**a**) Experimental outline. CFSE-labelled MD4 Ig-Tg B cells were pulsed *ex vivo* with DEL-OVA or DEL-OVA-Eα and co-transferred with CFSE-labelled naive control B cells into unimmunized mice. After various periods of time, splenocytes from recipient mice were analysed and transferred to secondary recipient mice which had received OTII Th cells and been immunized i.p. with OVA in Ribi 3 d before. (**b**) Gating strategy for proliferated and unproliferated Ig-Tg and control B cells. (**c,d**) Ratios of proliferated (**c**) and unproliferated (**d**) Ig-Tg B cells to control B cells in preimmunized (filled circles) and control unimmunized (open circles) secondary recipient mice, normalized to the ratio of Ig-Tg to control B cells at the time of transfer. Data shown as mean ± s.e.m. $n = 5$ independent experiments. (**e**) B-cell receptivity to T-cell help calculated as $(1 - F_I/F_C)$, where $F_I$ and $F_C$ are ratios of unproliferated Ig-Tg B cells to control cells in immunized and control secondary recipient mice, respectively. Data shown as mean ± s.e.m.

Ig-Tg B cells and cognate Th cells are above or within the reported physiologic frequencies of antigen-specific endogenous cells[27,28,34]. Based on these data, we conclude that continuous or recurrent exposure to antigen is not required for B cells' ability to acquire T-cell help and proliferate or to be recruited into GC, memory and PC responses *in vivo*. Moreover, B cells are recruited over a very broad range of antigen amounts acquired and level of antigenic peptides presented (Figs 2a–d and 4l,m). In contrast, timely availability of T-cell help is critical for recruitment of transiently antigen-primed B cells into the humoural response.

Our studies demonstrate that B cells transiently exposed to moderately crosslinking antigens have a limited window of time *in vivo* to acquire T-cell help and be recruited into the B-cell response. Quantitative analysis suggests that B cells' ability to acquire T-cell help and undergo proliferation *in vivo* decreases twofold within 24 h of transiently acquiring of antigen, and is completely abolished between 36 and 48 h. We show that *in vivo*, transiently antigen-primed B cells that do not get T-cell help undergo sequential changes in localization and expression of surface markers that are consistent with their coordinated inactivation (Fig. 7). At 12 h following transient exposure to

antigen, B cells localize to the border between the follicle and T-cell zone and express high levels of costimulatory molecules and antigen-derived peptides, which are downregulated after B cells begin to migrate back into follicles by 24 h.

Our studies also show that transiently antigen-experienced B cells are not rendered unresponsive to restimulation with antigen following functional and phenotypic inactivation. Transiently pulsed and then inactivated B cells have similar capacity as naive B cells to join GC and PC responses *in vivo* upon re-exposure to antigen and T-cell help. These data raise the possibility that B cells may go through multiple rounds of transient antigen acquisition followed by inactivation while waiting for T-cell help.

Surprisingly, we found that in the absence of productive T-cell help, B cells transiently primed with saturating amounts of low to moderately crosslinking antigen do not undergo apoptosis *in vivo*, in contrast to their rapid death *ex vivo*[20]. Of note, transient exposure to very highly multivalent particles does lead to B-cell loss *in vivo*, consistent with the previously reported negative impact of very strong BCR signalling on B-cell survival[23]. While tonic CD40L signalling by non-cognate Th cells promotes survival of anergic B cells *in vivo*[35], our studies

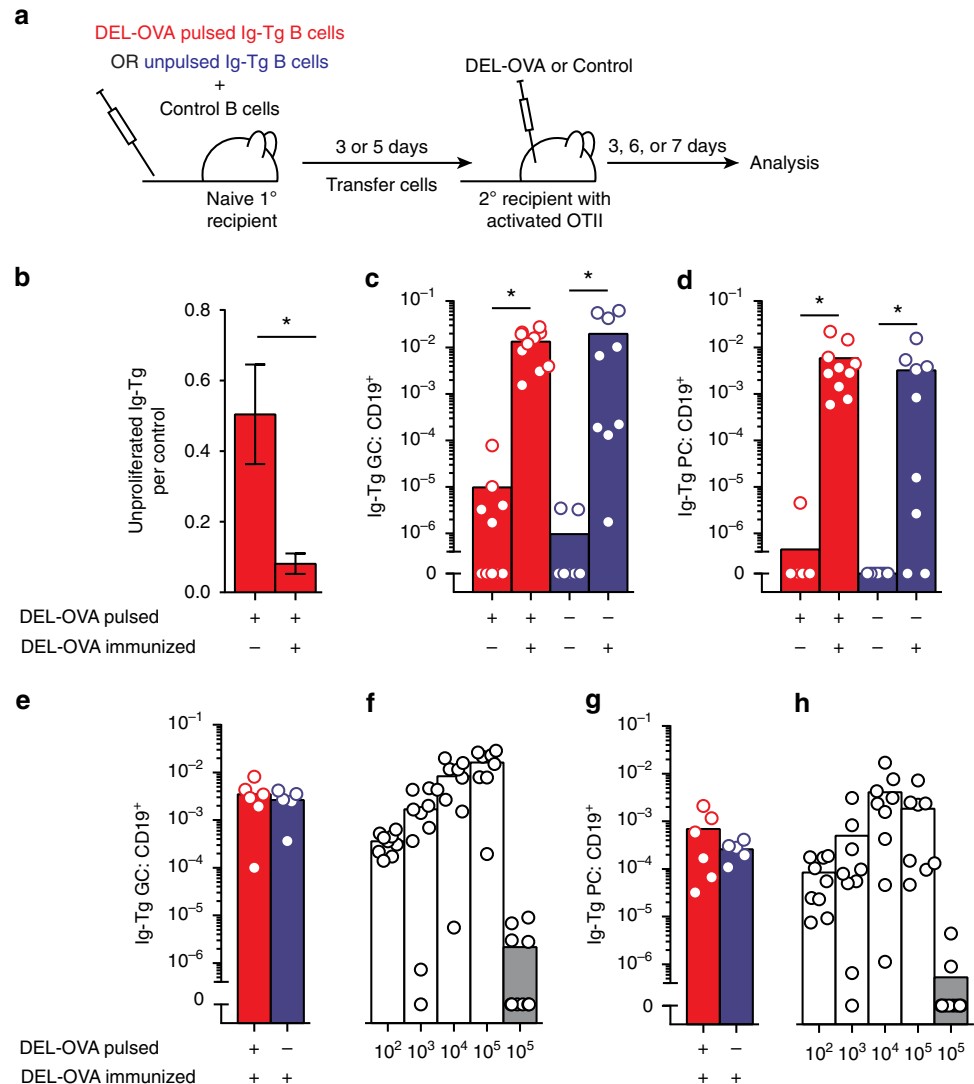

**Figure 6 | Inactivated B cells can re-acquire antigen and participate in the T-dependent humoural immune response.** Analysis of proliferation (**b**), GC (**c,e,f**) and PC (**d,g,h**) responses mounted by DEL-OVA-pulsed and then inactivated or naive Ig-Tg B cells upon reacquisition of antigen. (**a**) Experimental design. DEL-OVA-pulsed (50 µg ml⁻¹) or naive MD4 Ig-Tg B cells were transferred into unimmunized mice. After 3 days (**b–d**) or 5 d (**e,g**) splenocytes (**b–d**) or FACS-sorted unproliferated Ig-Tg B cells (**e,g**) were transferred from primary to secondary recipient mice which had received OTII Th cells and been preimmunized i.p. with OVA in Ribi for 3 days. Immediately after Ig-Tg cell transfer, non-control secondary recipient mice were re-immunized with DEL-OVA in Ribi. (**b**) Ratios of unproliferated Ig-Tg B cells to control B cells in non-reimmunized or DEL-OVA re-immunized secondary recipient mice, normalized to the input B-cell ratios from primary recipients. Data from n = 3 independent experiments with five mice per condition, shown as mean ± s.e.m. *P < 0.05 (two-tailed Mann–Whitney test). (**c–e,g**) Participation in the GC (**c,e**) and PC (**d,g**) response by antigen-pulsed and then inactivated (red) or unpulsed naive (blue) Ig-Tg B cells in DEL-OVA reimmunized or non-reimmunized secondary recipient mice 6 days (**c,d**) or 7 days (**e,g**) post re-immunization. (**f,h**) GC (**f**) and PC (**h**) response participation 7 d.p.t. of the indicated number of unpulsed naive Ig-Tg B cells to OVA-immunized recipient mice with OTII Th, either reimmunized with DEL-OVA (white) or non-reimmunized (grey). Ig-Tg B-cell fractions of total splenic B cells are shown; each dot represents one mouse, bars at mean. (**c,d**) Data from n = 5 independent experiments with 10 mice (DEL-OVA pulsed Ig-Tg) or n = 3 with nine mice (unpulsed Ig-Tg). (**e,g**) Data from n = 3 with 6 (pulsed Ig-Tg) or 5 (unpulsed Ig-Tg) mice. (**f,h**) Data from n = 2–3 independent experiments with 8–10 mice. *P < 0.05 (Kruskal–Wallis test with Dunn's post-test).

indicate that survival of B cells following transient antigen acquisition is independent of signals from cognate or non-cognate Th cells. Future studies should address which factors or combination of factors promote survival of transiently antigen-primed B cells *in vivo*.

In contrast to B cells transiently exposed to antigen, we detect progressive loss of antigen-specific B cells after 24 h of recurrent exposure to antigen in the absence of T-cell help *in vivo*. This is consistent with multiple studies demonstrating increased apoptosis and development of anergy in B cells continuously exposed to self-antigen[31,32]. While our findings challenge the postulate that antigen-experienced B cells must die in the absence of T-cell help[21], they are consistent with the 'time-honoured hypothesis' that predicts that tolerance is induced by prolonged antigen receptor signalling in the absence of a second signal[36]. Future studies should address the molecular mechanisms which direct mature B cells' fate decisions between tolerance and quiescence based on the duration of their exposure to antigen.

Based on the *in vivo* kinetic analysis of B-cell fate performed in this study, we speculate that duration of initial antigen acquisition

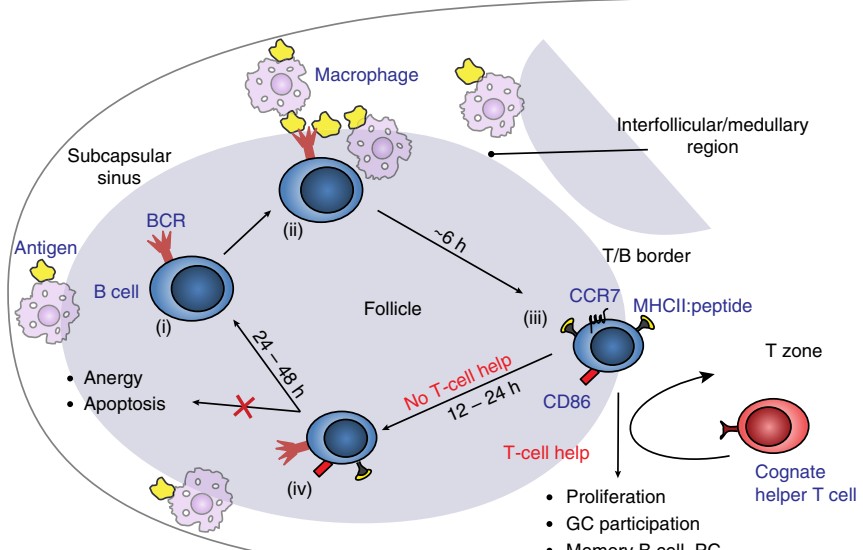

**Figure 7 | Proposed model of early antigen and T-cell help acquisition events by B cells.** The model summarizes the early events in large particulate antigen acquisition by B cells prior to generation of immune complexes and antigen transport onto FDC. Naive B cells in the follicle (i) migrate to the subcapsular sinus and interfollicular regions of secondary lymphoid structures or medullary sinuses, where they can transiently acquire antigen (ii) and then leave. Within 6 h, they localize to the T/B border (iii) where, if they encounter T-cell help within ∼12–24 h, they can proliferate and participate in the GC, memory and PC responses. If T-cell help is not acquired within this time, then B cells migrate back into the follicle and gradually downregulate expression of activation markers and antigen-derived peptide/MHCII complexes (iv). The inactivated B cells are not anergic and do not undergo apoptosis; rather they can acquire antigen again and have additional opportunities for recruitment into the T-dependent humoral immune response.

by B cells may be an important factor affecting the efficiency and clonal diversity of the B-cell response. Given the low frequency of antigen-specific Th cells during initiation of the primary immune response ($10^{-5}$–$10^{-6}$) (ref. 34) and the time it takes migratory dendritic cells to promote Th cell proliferation[37,38], it is not unlikely that productive encounters between antigen-exposed B cells and cognate Th cells may take a few days. Continuous exposure of B cells to antigen may then lead to their apoptosis or anergy prior to acquisition of T-cell help, limiting the number of B-cell clones recruited into the immune response. Therefore, when T-cell help is delayed, a transient mode of initial antigen acquisition (as observed for large particulate antigens deposited at restricted sites in secondary lymphoid organs) may favour survival of antigen-specific B cells *in vivo* and allow for subsequent opportunities to reacquire antigen and T-cell help and be recruited into the T-dependent humoral immune response (Fig. 7). Consistent with that, large particulate antigens, which are initially sequestered from B-cell follicles, promote more efficient humoral responses than small soluble antigens[39–41]. This is likely due to multiple factors, so future studies should investigate how the kinetics and localization of antigen acquisition by B cells affect the magnitude and breadth of the B-cell repertoire entering T-dependent humoral responses.

Altogether, our results indicate that transient antigen acquisition by B cells may be sufficient for their recruitment into T-dependent B-cell responses and that in contrast to continuous antigen exposure, transient acquisition of antigen does not always induce anergy or death in the absence of T-cell help. Such a mechanism may allow B cells multiple opportunities to be recruited into the humoral immune response and favour a more clonally diverse population of responding B cells. We suggest that the temporal dynamics of B-cell exposure to antigen are an important factor for consideration in vaccine development, particularly for immunocompromised people who have a limited number or diversity of responding T-cells[42–45].

## Methods

**Mice.** B6 (C57BL/6) mice were purchased from Charles River, NCI or the Jackson Laboratory. B6-CD45.1 (Ptprc$^a$ Pepc$^b$/BoyJ) and TCRα$^{-/-}$ ($Tcra^{tm1Mom}$) mice were purchased from the Jackson Laboratory. BCR transgenic (Ig-Tg) Hy10 (ref. 24) and MD4 mice[29] and TCR transgenic OTII mice[25] (all C57BL/6 background) were generously provided by Jason Cyster. Ig-Tg MD4 and Hy10 mice were crossed with B6-CD45.1 mice and maintained on this background. Hy10 B cells are capable of Ig class switching and were used for experiments characterizing the GC response (Figs 1 and 2). MD4 Ig-Tg mice have a more highly penetrant transgene, allowing for higher recovery of Ig-Tg B cells (∼95% of total B cells), and were used in all other experiments. Donor and recipient mice were 6–12 weeks of age. All mice were maintained in a specific pathogen-free environment and protocols were approved by the Institutional Animal Care and Use Committee of the University of Michigan.

**Antigen preparation.** Duck eggs were locally purchased and DEL was purified as previously described[24]. HEL and OVA were purchased from Sigma. DEL or HEL was conjugated to OVA via glutaraldehyde cross-linking as previously described[24]. For conjugation of DEL-OVA to an I-E alpha chain peptide containing the epitope Eα 52–68 (Eα peptide), Eα peptide with four amino acids from the native sequence flanking each end and a cysteine replacing the N-terminal phenylalanine to allow for maleimide targeting was purchased from GenScript (CAKF**ASFEAQGALANIAVDKA**NLDV, residues 52-68 bold). This peptide was crosslinked to DEL-OVA at ten fold molar excess using bismaleimidoethane (Pierce) according to the manufacturer's directions.

For generation of multivalent DEL (mDEL), DEL was combined with biotin-NHS (Fisher) at a 1:2 molar ratio and conjugated and purified according to the manufacturer's directions. Biotinylated DEL was incubated for 30 min on ice with purified streptavidin (Sigma) at a 10:1 molar ratio and the conjugate was purified with 40 kDa desalting columns (Bio-Rad) according to the manufacturer's directions.

For generation of DEL-coated microspheres (sphDEL), 0.11 μm streptavidin coated polystyrene microspheres (Bangs Laboratories) were dialysed into PBS and combined with a saturating amount of DEL-bio as described previously[33].

**Adoptive transfer and immunization.** Spleens were harvested from male donor OTII mice and pressed through 70 μm nylon cell strainers in DMEM supplemented with 4.5 g l$^{-1}$ glucose, L-glutamine and sodium pyruvate, 2% FBS, 10 mM HEPES, 50 IU ml$^{-1}$ of penicillin and 50 μg ml$^{-1}$ of streptomycin (HyClone), (DMEM 2%). Splenocytes were centrifuged for 7 min at 380$g$, 4 °C and resuspended in 0.14 M NH$_4$Cl in 0.017 M Tris buffer, pH 7.2 for erythrocyte lysis, washed twice with DMEM 2% and counted using a Cellometer Auto X4 (Nexcelom). The fraction of CD19$^-$ CD8$^-$ CD4$^+$ Vβ5$^+$ (OTII) splenocytes was

determined by flow cytometry, and $5 \times 10^5$ OTII cells were transferred intravenously (i.v.) to male B6 recipient mice, which had been recently purchased or bred in-house.

Ig-Tg and wild-type B cells were enriched from male and female donor mice by negative selection as previously described[46]. For labelling with carboxyfluorescein succinimidyl ester (CFSE), purified Ig-Tg or wild-type B cells were washed and resuspended in DMEM supplemented as above, but with 1% FBS at fewer than $10^7$ cells per ml. CFSE was added to the cells at a final concentration of 1 μM, and cells were incubated for 20 min at 37 °C. Two millilitres of FBS was layered under the cells, which were centrifuged for 7 min at 380$g$, 4 °C and resuspended in DMEM 2%.

For transient exposure to antigen, purified Ig-Tg B cells were incubated with HEL, HEL-OVA, DEL-OVA, DEL-OVA-Eα, mDEL or sphDEL ex vivo for 5 min at 37 °C, washed four times with room temperature DMEM 2%, mixed with CFSE-labelled naive control cells where indicated, and transferred i.v. to recipient mice or cultured in DMEM supplemented as above with 10% FBS and 50 μM β-mercaptoethanol.

Where indicated, recipient mice were immunized intraperitoneally (i.p.) with 50 μg OVA or DEL-OVA in Ribi (Sigma), or subcutaneously with 50 μg OVA, HEL-OVA or DEL-OVA emulsified in CFA (Sigma), prepared according to the manufacturer's directions.

For transfer of Ig-Tg and control cells from primary (1°) to secondary (2°) recipient mice, single cell suspensions of splenocytes from 1° recipients were made and treated for erythrocyte lysis as above, resuspended in DMEM 2%, counted and transferred i.v. to 2° recipients. At least 2 million splenocytes from 1° recipients were reserved for flow cytometry staining to determine the ratio of Ig-Tg to control cells. For experiments in which unproliferated antigen-pulsed and naive Ig-Tg B cells were sorted from 1° recipients prior to transfer into 2° recipients (Fig. 6e,g), splenocytes from 1° recipients were either stained directly for sorting or depleted of non-B cells by negative selection as above, then stained for sorting with fluorochrome-conjugated CD19, CD4, CD8, CD45.1 and CD45.2 diluted in PBS supplemented with 0.5% FBS, 2 mM EDTA and 10 mM HEPES (sorting buffer) for 20 min on ice. Cells were washed twice and resuspended in sorting buffer. CD19$^+$ CD4$^-$ CD8$^-$ CD45.1$^+$ CD45.2$^-$ CFSE$^{high}$ Ig-Tg B cells and CD19$^+$ CD4$^-$ CD8$^-$ CD45.1$^-$ CD45.2$^+$ CFSE$^{high}$ control B cells were sorted on a FACSAria and resuspended in DMEM 2%. Within each experiment, the same number of inactivated or naive B cells ($4-7 \times 10^3$) were transferred i.v. to 2° recipients, which were re-immunized with 50 μg DEL-OVA in Ribi, i.p.

**Flow cytometery.** Single-cell suspensions from spleens or lymph nodes were incubated with biotinylated antibodies (Supplementary Table 1) for 20 min on ice, washed twice with 200 μl PBS supplemented with 2% FBS, 1 mM EDTA and 0.1% NaN$_3$ (FACS buffer), incubated with fluorophore-conjugated antibodies and streptavidin (Supplementary Table 1) for 20 min on ice, washed twice more with 200 μl FACS buffer, and resuspended in FACS buffer for acquisition. For intracellular staining, surface-stained cells were fixed and permeabilized for 20 min on ice with BD Cytofix/Cytoperm buffer, washed twice with 200 μl BD Perm/Wash buffer, incubated with Alexa 647-conjugated HEL for 20 min on ice, followed by two washes with 200 μl Perm/Wash buffer, and resuspended in FACS buffer for acquisition. Data were acquired on a FACSCanto or LSRFortessa and analysed using FlowJo (TreeStar).

**ELISAs.** Serum was collected from recipient mice serially bled through the saphenous or tail vein or terminally bled through the portal vein by incubating blood for 20 min at room temperature in polypropylene microcentrifuge tubes followed by centrifugation for 10 min at 2,000$g$, 4 °C. Supernatant was collected and stored at − 20 °C until assayed. 96-well polyvinylchloride plates were coated overnight at room temperature in a humid chamber with 100 μl · 10 mg/ml HEL in 22 mM sodium carbonate, 34 mM sodium bicarbonate buffer, 0.2% MgCl$_2$, pH 9.8. The wells were then blocked for 4 h with 100 μl 10% FBS, 0.04% NaN$_3$ in PBS at room temperature in a humid chamber and washed once with 0.05% Tween 20 in PBS (PBST). Fivefold serial dilutions of sera in duplicate were added to the wells with threefold dilutions of appropriate standards (pooled sera from MD4 mice and purified HyHEL10 IgG$_1$ for IgM$^a$ and IgG$_1^a$, respectively) and the plates were incubated for 2 h at room temperature in a humid chamber. Wells were washed four times with 200 μl PBST and 100 μl biotinylated IgM$^a$ and IgG$_1^a$ antibodies diluted 1:4,000 and 1:1,000, respectively, in 1% BSA, 0.02% MgCl$_2$, 0.02% NaN$_3$ Tris buffer, pH 8 were used to detect bound serum antibody. Plates were incubated overnight at room temperature in a humid chamber and washed four times with 200 μl PBST. Alkaline phosphatase-conjugated streptavidin was diluted 1:4,000 in Tris buffer, and 100 μl was added to the wells. Plates were incubated 2 h and washed four times with 200 μl PBST. Hundred microlitres of 1 mg ml$^{-1}$ $p$-nitrophenyl phosphate in carbonate buffer was added to the wells, and absorbance at 405 nm was measured with a Synergy HT microplate reader (BioTek Instruments). The concentrations of anti-HEL antibodies were calculated from standard curves generated from a stock of pooled serum from naive MD4 mice for IgM$^a$ and purified HyHEL10 antibody for IgG$_1^a$. Significant differences were determined by unpaired, two-tailed Student's $t$-test in Prism (GraphPad).

**Immunofluorescence.** Spleens and ILNs were harvested from recipient mice, fixed for 1 h in 1% paraformaldehyde in PBS on ice, washed with PBS, blocked overnight in 30% sucrose, 0.1% NaN$_3$ in PBS, embedded in Tissue-Tek optimum cutting temperature compound, snap-frozen in dry ice and ethanol, and stored at − 70 °C. Thirty micron cryostat sections were cut from the tissue blocks, affixed to Superfrost Plus microscope slides (Fisher) and stained with IgD, CD45.1 and Bcl6 or CD3, CD19 and CD45.1 as previously described[2]. Slides were analysed at room temperature using a Leica SP5 with argon and helium-neon lasers, 2-channel Leica SP spectral fluorescent PMT detector, and a 20 × oil-immersion objective with a numerical aperture of 0.7. Images were processed using Imaris (Bitplane) and analysed using ImageJ (NIH) and Matlab (Mathworks). For B cell localization analysis, the average normalized distance of Ig-Tg cells to the T/B border in a follicle was defined as the average distance to the T/B border of the Ig-Tg B cells in the follicle divided by the average distance to the T/B border of points randomly distributed within the follicle, such that zero corresponds to cell distribution along the T/B border and one to random distribution within the follicle.

**Statistics.** Statistical tests were chosen in consultation with the University of Michigan Center for Statistical Consultation and Research and performed as indicated using Prism 6 (GraphPad). Differences between groups not annotated by an asterisk did not reach statistical significance. Power calculations to determine sample size for experiments making statistical comparisons between groups were performed after initial experiments enabled estimation of effect size. No blinding or randomization was performed for animal experiments, and no animals or samples were excluded from analysis.

**Data availability.** The data supporting the findings of this study are available within the article and its Supplementary Information files, or are available from the corresponding author upon reasonable request.

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

## Acknowledgements

We thank W. Dunnick for discussions, J. Cyster for provision of mice, T. Phan and K. Choudhuri for critical reading of the manuscript, and the University of Michigan flow cytometry core for cell sorting assistance. Supported by the National Institute of Health (R01 AI106806) to I.G.

## Author contributions

J.S.T. designed and performed the experiments, interpreted the results and prepared the manuscript. M.M. performed confocal microscopy. Z.L.B. designed the experimental procedures, wrote the image analysis software and assisted with *in vivo* experiments. I.G. designed and performed the experiments, interpreted the results, supervised the research and prepared the manuscript.

## Additional information

**Competing interests:** The authors declare no competing financial interests.

