## [Peer Review File · Nature Communications]

Reviewers' comments:

Reviewer #1 (Remarks to the Author):

This manuscript presents a thorough and careful analysis of the in vivo consequences of transient exposure of B cells to antigen. In general the experiments are well performed and interpreted and cohesive and justifiable arguments are presented. The case is clearly made that B cells can enter T-dependent responses following transient antigen exposure or persist in a functional (non-nergic) form if T cell help is not received.

My one major criticism of this study is that the issue of antigen valency was not addressed. This, whilst the situation is carefully analyzed in the case of a soluble and presumably paucivalent antigen (HEL/DEL-OVA), it is not clear that the same persistence of transiently antigen-exposed B cells would occur if the B cells had undergone stronger cross-linking of their BCRs. Do the authors have any evidence that could be presented along these lines? Since the HEL system is employed here, the alternate fates of B cells that bind soluble versus membrane bound HEL self-antigen suggests that the intensity of BCR cross-linking may have a significant effect on the fate of the B cells following Ag contact. Some data to address this issue would greatly strengthen the significance of this study.

Other points:

- 1) In Fig 3b it is not clear from the graph labels what is being expressed here - presumably "Live B cells: Input B cells" would be a more accurate label for the y-axis?
- 2) Fig. 3d refers to "unproliferated" Ig-Tg cells. Are there proliferating cells? eg in the B6 recipients (due to endogenous T cell help) and, if so, how is this result to be clearly interpreted?
- 3) It would be helpful if a larger and brighter version of Fig. 1c was utilised.
- 4) Similarly for Fig 4a, b it is very difficult to determine where the T:B border is.
- 5) There appear to be columns missing in Figs 6d, e.
- 6) Reference 32 is incomplete.

Reviewer #2 (Remarks to the Author):

It has long been held as fact that B-cells, once activated by antigen, are destined to follow one of three pathways. The B cells could proliferate and differentiate as part of a productive immune response; they could assume an anergic state from which it would be very difficult to rouse them, or they could undergo programmed cell death. The interesting aspect of the work from Taylor et al is that they have tested this truism with a simple but clever system and, perhaps not surprisingly, found it wanting. They report with striking clarity that an antigen-activated B-cell that fails to receive either T-cell help or additional exposure to antigen, resumes its naïve state, including the capacity to be recruited into a productive immune response upon exposure to the antigen in the presence of help.

There are three situations tested in the transgenic B and T cell transfer systems employed here.

1. B cells transiently exposed to antigen but provided with unlimited T cell were able to enter into the immune response, producing plasma cells, germinal centers and memory B cells. There was a relationship between dose of antigen and the proportional representation of the different fates – more antigen associated with more PC, for example – but even doses of antigen that were sub-

activating ex vivo were able to support proliferation and differentiation in vivo with T cell help.

2. B cell transiently exposed to antigen in vitro, transferred in vivo but now in the absence of T cell help and the absence of cognate antigen, persisted in the spleen over the time period of the experiment (3d) and resumed their naïve, un-activated phenotype. Interestingly, the B-cells showed a diminishing capacity to respond to T-cell help if it was provided in delayed fashion over this same time period of 3d. If the recipient had been immunised with the B-cell antigen at the time of transfer, providing ongoing exposure to the B cell antigen in the absence of T-cell help, then the B cells were lost from the spleen.

3. Finally, these restored naïve B cells were capable of being recruited into an immune response with kinetics and magnitude equal to that of true naïve B-cells, if they were exposed to their cognate antigen in vivo with T-cell help. Again, this was tested at 3 days.

Positives:

Very clear results and well presented. An elegant system that is very well constructed to examine the key questions but with sufficient depth to answer most obvious criticisms and to also add additional information to the report. Basic result will be of considerable interest to the field.

Limitations:

Time frame of the key experiment is somewhat short. Transfers of antigen pulsed B cells are done 3 or 5 days after first transfer in the experiments shown in Fig 6. What was the frequency of 'returned to naïve' B-cells at that time, day 5? Is it possible that only very few antigen-reactive B cells would need to be present in this system to give the observed results following re-immunization? That is, how quantitative a comparison is it between the 'returned to naïve' state at day 3 or 5 and naïve B-cells? Especially since the cells are being put into an already primed mouse (OVA in Ribl) and then boosted immediately. Huge amounts of T cell help, probably, available. That is, is there any idea of the dynamic range of the assays being used to measure precursors. The secondary transfer population in Fig 6 may need to contain only a small number of residual antigen activated B cells from the first exposure to generate the response seen.

Questions:

Methods state "the average normalized distance of Ig-Tg cells to the T/B border in a follicle was defined as the average distance to the T/B border of the Ig-Tg B cells in the follicle divided by the average distance to the T/B border of points randomly distributed within the follicle, such that 1 corresponds to cell distribution along the T/B border, and 0 to random distribution within the follicle." I might have this wrong, but shouldn't this be the other way round for interpreting 0 and 1? Looks like Figure 4 legend description differs from methods as well.

How long lasting is the return to naïve state? The tracking by frequency experiments last 72 hours and show what looks like continuous decline of both stimulated and unstimulated transferred B cells. The problem is that at later times, one could have plateaued (control) and the other continued its decline, creating an ever increasing gap. Could this be extended? Is it also possible that the stimulated B cells preferentially reside in the spleen and don't recirculate, at least compared to the control B cells, giving an artificially high ratio as the control population is equilibrated throughout the mouse. A more detailed explanation of how these values were calculated would certainly help.

Figure 3 legend. (e) mislabeled as (d).

I am somewhat confused by the data in Fig 5c and d. The value in (c) of 0.15 approximately for proliferated IgTg:control and (d) 0.6 for the ratio of unproliferated Ig-Tg:Control at 12 hours in control unimmunized mice is hard to understand. Does it mean in (c) that there were more proliferating cells on the control injected B cells than in the IgTg injected B cells? Does it mean in (d) that there were more unproliferated cell in the control injected B cells than in the injected IgTg B cells? If that is the case, can both of these be true or have I misunderstood the experiment or the measurement? Perhaps this could be clarified?

We would like to thank both reviewers for the careful reading of the manuscript, meticulous analysis, and excellent suggestions! We thought that all suggested points were very important and attempted to address all of them in the new version of the manuscript resubmitted below.

We have added the following Figures and Supplementary Figures to the manuscript: **Fig. 3f, Fig. 6f, g, Supplementary Fig. 3c, d, and Supplementary Fig. 4.**

We also slightly modified the manuscript's discussion and conclusions to extend them to the new data.

Finally, in addition to addressing the major points experimentally, we attempted to clarify the minor questions/imprecisions noticed and asked by the reviewers in the figures and the text of the manuscript. All changes to the text have been highlighted.

Reviewers' comments:

Reviewer #1 (Remarks to the Author):

This manuscript presents a thorough and careful analysis of the in vivo consequences of transient exposure of B cells to antigen. In general the experiments are well performed and interpreted and cohesive and justifiable arguments are presented. The case is clearly made that B cells can enter T-dependent responses following transient antigen exposure or persist in a functional (non-anergic) form if T cell help is not received.

My one major criticism of this study is that the issue of antigen valency was not addressed. This, whilst the situation is carefully analyzed in the case of a soluble and presumably paucivalent antigen (HEL/DEL-OVA), it is not clear that the same persistence of transiently antigen-exposed B cells would occur if the B cells had undergone stronger cross-linking of their BCRs. Do the authors have any evidence that could be presented along these lines? Since the HEL system is employed here, the alternate fates of B cells that bind soluble versus membrane bound HEL self-antigen suggests that the intensity of BCR cross-linking may have a significant effect on the fate of the B cells following Ag contact. Some data to address this issue would greatly strengthen the significance of this study.

We agree that the effect of Ag valency is a very important question to address and thank the reviewer for bringing it to our attention. To address it, we generated additional multivalent Ags, streptavidin – bio-DEL (with an expected valency of 4) and very highly multivalent DEL-coated polystyrene beads. We then compared these Ags and DEL-OVA with soluble HEL and DEL with respect to their Ig-Tg BCR binding and cross-linking/internalization abilities (**Supplementary Fig. 2c, d**). Next we monitored B cell survival in vivo for 7 days following transient acquisition of saturating amounts of soluble, moderately multivalent, and highly multivalent Ags, and found that transient exposure to soluble (HEL) or moderately multivalent Ags (SA-DEL or DEL-OVA) did not result in B cell disappearance from the secondary lymphoid organs within this time frame, but transient exposure to very highly multivalent Ag had a negative effect on B cell survival, a trend observed as early as 12 hours post transfer, with a significant decline in B cell number by 7 days (**Fig. 3f, Supplementary Fig. 2e**).

Other points:

1) In Fig 3b it is not clear from the graph labels what is being expressed here - presumably "Live B cells: Input B cells" would be a more accurate label for the y-axis?

This axis label has been updated.

2) Fig. 3d refers to "unproliferated" Ig-Tg cells. Are there proliferating cells? eg in the B6 recipients (due to endogenous T cell help) and, if so, how is this result to be clearly interpreted?

Proliferation of a small fraction (at most 7%) of DEL-OVA-primed Ig-Tg B cells was observed in both B6 and TCR $\alpha^{-/-}$ recipient mice, indicating that proliferation was T-independent. We used the unproliferated fraction of Ig-Tg B cells for quantification of survival to avoid the confounding effects of proliferation.

3) It would be helpful if a larger and brighter version of Fig. 1c was utilised.

4) Similarly for Fig 4a, b it is very difficult to determine where the T:B border is.

The size and brightness of the confocal micrographs have been increased.

5) There appear to be columns missing in Figs 6d, e.

The missing columns have been added back.

6) Reference 32 is incomplete.

This reference has been updated.

Reviewer #2 (Remarks to the Author):

It has long been held as fact that B-cells, once activated by antigen, are destined to follow one of three pathways. The B cells could proliferate and differentiate as part of a productive immune response; they could assume an anergic state from which it would be very difficult to rouse them, or they could undergo programmed cell death. The interesting aspect of the work from Taylor et al is that they have tested this truism with a simple but clever system and, perhaps not surprisingly, found it wanting. They report with striking clarity that an antigen-activated B-cell that fails to receive either T-cell help or additional exposure to antigen, resumes its naïve state, including the capacity to be recruited into a productive immune response upon exposure to the antigen in the presence of help.

There are three situations tested in the transgenic B and T cell transfer systems employed here.

1. B cells transiently exposed to antigen but provided with unlimited T cell were able to enter into the immune response, producing plasma cells, germinal centers and memory B cells. There was a relationship between dose of antigen and the proportional representation of the different fates – more antigen associated with more PC, for example – but even doses of antigen that were sub-activating ex vivo were able to support proliferation and differentiation in vivo with T cell help.

2. B cell transiently exposed to antigen in vitro, transferred in vivo but now in the absence of T cell help and the absence of cognate antigen, persisted in the spleen over the time period of the experiment (3d) and resumed their naïve, un-activated phenotype. Interestingly, the B-cells showed a diminishing capacity to respond to T-cell help if it was provided in delayed fashion over this same time period of 3d. If the recipient had been immunised with the B-cell antigen at the time of transfer, providing ongoing exposure to the B cell antigen in the absence of T-cell help, then the B cells were lost from the spleen.

3. Finally, these restored naïve B cells were capable of being recruited into an immune response with kinetics and magnitude equal to that of true naïve B-cells, if they were exposed to their cognate antigen in vivo with T-cell help. Again, this was tested at 3 days.

Positives:

Very clear results and well presented. An elegant system that is very well constructed to examine the key

questions but with sufficient depth to answer most obvious criticisms and to also add additional information to the report. Basic result will be of considerable interest to the field.

Limitations:

Time frame of the key experiment is somewhat short. Transfers of antigen pulsed B cells are done 3 or 5 days after first transfer in the experiments shown in Fig 6. What was the frequency of ‘returned to naïve’ B-cells at that time, day 5? Is it possible that only very few antigen-reactive B cells would need to be present in this system to give the observed results following re-immunization? That is, how quantitative a comparison is it between the ‘returned to naïve’ state at day 3 or 5 and naïve B-cells? Especially since the cells are being put into an already primed mouse (OVA in Rib) and then boosted immediately. Huge amounts of T cell help, probably, available. That is, is there any idea of the dynamic range of the assays being used to measure precursors. The secondary transfer population in Fig 6 may need to contain only a small number of residual antigen activated B cells from the first exposure to generate the response seen.

This is a very important point and we thank the reviewer for bringing it to our attention. We have attempted to address it by performing a titration of naïve Ig-Tg cells to establish the dynamic range of the response (Fig. 6f, h), and transferring a number of Ag-primed and then inactivated Ig-Tg B cells that fell within this range. We sorted and transferred $3-7 \times 10^3$ unproliferated, Ag-primed and then inactivated or naïve Ig-Tg cells from primary recipient mice 5 days post transfer and found that they generated comparable GC and PC responses (Fig. 6e, g). Although it is not the only marker that could indicate a residually activated state, CD86 expression was assessed at the time of sorting and we found no more than 7% of the Ag-experienced population of Ig-Tg cells expressed slightly higher amounts of CD86 than the naïve cotransferred control cells (Supplementary Fig. 4b).

Questions:

Methods state “the average normalized distance of Ig-Tg cells to the T/B border in a follicle was defined as the average distance to the T/B border of the Ig-Tg B cells in the follicle divided by the average distance to the T/B border of points randomly distributed within the follicle, such that 1 corresponds to cell distribution along the T/B border, and 0 to random distribution within the follicle.” I might have this wrong, but shouldn’t this be the other way round for interpreting 0 and 1? Looks like Figure 4 legend description differs from methods as well.

It had been inadvertently reversed in the methods and has been corrected.

How long lasting is the return to naïve state? The tracking by frequency experiments last 72 hours and show what looks like continuous decline of both stimulated and unstimulated transferred B cells. The problem is that at later times, one could have plateaued (control) and the other continued its decline, creating an ever increasing gap. Could this be extended? Is it also possible that the stimulated B cells preferentially reside in the spleen and don’t recirculate, at least compared to the control B cells, giving an artificially high ratio as the control population is equilibrated throughout the mouse. A more detailed explanation of how these values were calculated would certainly help.

We sought to address these points by extending the time course to 7 days and collecting axil, brachial, cervical, inguinal, and mesenteric LNs in addition to spleens. Consistent with our earlier results, we found that transient exposure to soluble or moderately multivalent Ags did not result in B cell disappearance from the secondary lymphoid organs within this time frame (Fig. 3f). Both Ig-Tg B cells and cotransferred control cells recirculated through the LNs, with the control cells appearing there earlier, as the reviewer predicts. For clarity, the data is presented as the sum of Ig-Tg B cells remaining in the combined LNs and spleen. We have also attempted to clarify how these values are calculated in the text and figure legends.

Figure 3 legend. (e) mislabeled as (d).

The legend has been updated.

I am somewhat confused by the data in Fig 5c and d. The value in (c) of 0.15 approximately for proliferated IgTg:control and (d) 0.6 for the ratio of unproliferated Ig-Tg:Control at 12 hours in control unimmunized mice is hard to understand. Does it mean in (c) that there were more proliferating cells on the control injected B cells than in the IgTg injected B cells? Does it mean in (d) that there were more unproliferated cell in the control injected B cells than in the injected IgTg B cells? If that is the case, can both of these be true or have I misunderstood the experiment or the measurement? Perhaps this could be clarified?

Both the proliferated and unproliferated Ig-Tg B cells are normalized to the same population of naïve (non-proliferating) co-transferred control cells. To compare Ig-Tg proliferation among different mice and experiments, the ratios of proliferated and unproliferated Ig-Tg to control cells are normalized to the ratio of Ig-Tg to control cells at the time of the transfer from primary to secondary recipient mice. We attempted to clarify this by modifying the axis labels for **Figure 5c, d** and additional description in the text.

REVIEWERS' COMMENTS:

Reviewer #1 (Remarks to the Author):

The authors have responded satisfactorily to all my initial concerns.

Reviewer #2 (Remarks to the Author):

The revisions made by Turner et al have substantially improved the work, corrected errors and addressed concerns about clarity and interpretation. I am happy to recommend acceptance of the manuscript and have only two very minor points for consideration.

- 1) I noticed in the legend to Fig 3 that "data" is used in the singular rather than plural. This may be in other places so should be checked throughout.
- 2) the added reference (No 33) lacks the doi information that all other references contain.

We would like to thank both reviewers again for their careful and thorough review of the manuscript. We have attempted to address all concerns in the new version of the manuscript resubmitted below.

All changes to the text have been highlighted.

Reviewers' comments:

The revisions made by Turner et al have substantially improved the work, corrected errors and addressed concerns about clarity and interpretation. I am happy to recommend acceptance of the manuscript and have only two very minor points for consideration.

1) I noticed in the legend to Fig 3 that “data” is used in the singular rather than plural. This may be in other places so should be checked throughout.

This has been altered to the plural, along with several other instances of this error.

2) the added reference (No 33) lacks the doi information that all other references contain.

Doi information has been added for this reference.